# Estimating Health Adjusted Age at Death (HAAD)

**Kjell Arne Johansson[1,2], Jan-Magnus Økland[1], Eirin Krüger Skaftun[1], Gene Bukhman[3], Ole Frithjof Norheim[1,4], Matthew M. Coates[3], Øystein Ariansen Haaland[1]***

1 Department of Global Public Health and Primary Care, Bergen Centre for Ethics and Priority Setting (BCEPS), University of Bergen, Bergen, Norway, 2 Department of Addiction Medicine, Haukeland University Hospital, Bergen, Norway, 3 Program in Global NCDs and Social Change, Harvard Medical School, Boston, Massachusetts, United States of America, 4 Harvard T. H. Chan School of Public Health, Harvard University, Boston, Massachusetts, United States of America

* oystein.haaland@uib.no

**Data Availability Statement:** The data used to create the tables and figures in this paper can be accessed without restrictions at https://doi.org/10.5281/zenodo.3258330.

## Abstract

### Objectives

At any point in time, a person's lifetime health is the number of healthy life years they are expected to experience during their lifetime. In this article we propose an equity-relevant health metric, Health Adjusted Age at Death (HAAD), that facilitates comparison of lifetime health for individuals at the onset of different medical conditions, and allows for the assessment of which patient groups are worse off. A method for estimating HAAD is presented, and we use this method to rank four conditions in six countries according to several criteria of "worse off" as a proof of concept.

### Methods

For individuals with specific conditions HAAD consists of two components: past health (before disease onset) and future expected health (after disease onset). Four conditions (acute myeloid leukemia (AML), acute lymphoid leukemia (ALL), schizophrenia, and epilepsy) are analysed in six countries (Ethiopia, Haiti, China, Mexico, United States and Japan). Data from 2017 for all countries and for all diseases were obtained from the Global Burden of Disease Study database. In order to assess who are the worse off, we focus on four measures: the proportion of affected individuals who are expected to have HAAD<20 (T20), the 25th and 75th percentiles of HAAD for affected individuals (Q1 and Q3, respectively), and the average HAAD (aHAAD) across all affected individuals.

### Results

Even in settings where aHAAD is similar for two conditions, other measures may vary. One example is AML (aHAAD = 59.3, T20 = 2.0%, Q3-Q1 = 14.8) and ALL (58.4, T20 = 4.6%, Q3-Q1 = 21.8) in the US. Many illnesses, such as epilepsy, are associated with more lifetime health in high-income settings (Q1 in Japan = 59.2) than in low-income settings (Q1 in Ethiopia = 26.3).

**Funding:** Funding to was received from Bill & Melinda Gates Foundation through the Disease Control Priorities Ethiopia (DCP-Ethiopia) project grant to the University of Bergen and Harvard T.H. Chan School of Public Health (grant number: OPP1162384) (authors: KAJ, JMØ, EKS, MMC).

**Competing interests:** The authors have declared that no competing interests exist.

**Abbreviations:** AD, age at death; ALL, acute lymphoid leukemia; AML, acute myeloid leukemia; CEA, cost-effectiveness analysis; DALY, disability-adjusted life-years; GBD, Global Burden of Diseases; HAA, health adjusted age; HAAD, Health adjusted age at death; HALE, healthy life expectancy; HALY, healthy life years; PID, Period of Increased Disability; PIM, Period of Increased Mortality; YLD, years lived with disability; YLL, years of life lost.

## Conclusion

Using HAAD we may estimate the distribution of lifetime health of all individuals in a population, and this distribution can be incorporated as an equity consideration in setting priorities for health interventions.

## Introduction

All health systems have budget constraints and limited resources. Methods for health economic evaluations, like cost-effectiveness analysis (CEA), are essential in health policy and are extensively used to rank health services by their expected efficiency [1]. However, few people endorse strict health maximisation [2], and fairness criteria may be included in such rankings [3, 4]. For example, one may give higher priority to interventions that target those with the most severe illnesses [5–7], especially in relation to decisions about the pricing and reimbursement of new medicines and devices [8, 9]. Policy makers in countries like Norway [10] and the Netherlands [11] have already started using severity measurement methods.

In this paper, the terms "illness", "disease" and "condition" are used interchangeably, and include all adverse medical conditions, such as injuries, syndromes, birth defects, and infections. The term "severity of illness" involves both substantial value disagreements and a wide range of interpretations [12, 13]. To sidestep misunderstandings, we use the concept of "health status at disease onset" rather than severity of illness. Three perspectives on how to measure health status dominate in the literature. One view considers current health [14], one considers health over future years [15], and one considers health over the lifetime [16–18]. In this paper we conform to the last and focus on the lifetime health that individuals with a particular illness are expected to achieve before they die [18]. Technically, only the future health (after disease onset) is affected directly by the disease, but lifetime health is the sum of past (before disease onset) and future health expectancies. These differences in years of healthy life lived before disease onset among people with different diseases also inform the potential lifetime health that can be attained and are part of measuring health status by the view we use.

Clinical definitions of severity of illness often include urgency, but our definition of health status does not. Urgency pertains to the timing of treatment and how this influences the prognosis of a condition. Conditions that are severe from a lifetime health perspective, like multiple sclerosis in young patients, do not necessarily require urgent interventions. To underscore that we use a lifetime health perspective when measuring health status, we will from now on use the term "lifetime health" instead of "health status".

The Global Burden of Disease (GBD) study provides critical summary measures of population health that are relevant when evaluating and comparing health systems [3]. These measures include disability-adjusted life years (DALY) and health adjusted life expectancy (HALE). GBD uses a prevalence-based approach, where DALYs are calculated for a set of diseases by summing the years of life lost (YLL) compared to a reference life expectancy and years lived with disability (YLD) in one particular year due to each disease [19]. For a particular condition and a particular year, YLL is the sum of all the years lost for the individuals who died from the condition during that year. The reference is the age-adjusted life expectancy (LE) from a life table derived from the mortality rates in the locations with the lowest age-specific mortality in the GBD study [20]. YLD, on the other hand, is the sum of the health loss due to the condition during the year across people living with the condition [21]. DALYs aggregated from YLLs and YLDs are a measure of overall population burden. HALE measures the life

expectancy in a population, adjusting for the disability experienced in the population, using age-specific mortality rates and YLDs *per capita* [22]. A major limitation of these measures is that they do not capture how the condition affects the distribution of lifetime health at disease onset across individuals in the population.

We propose a framework where this distribution is an integral part. A key component in this framework is the new metric Health Adjusted Age at Death (HAAD). In this paper, we present a method for estimating HAAD, and show how to use the estimated HAAD to rank conditions at disease onset. We consider four conditions and six countries to illustrate how and why our framework is relevant for priority setting in health care and the measurement of population health.

## Methods

### Definition of HAAD

HAAD measures lifetime health for individuals with specific conditions and consists of two components: past health and future expected health. We obviously do not know the actual time of death for people dying in the future, but we do have some knowledge about the expected distribution across individuals. Consider, for example, two people aged 30 (Ann) and 50 (Bob) who each get a disease. The prognosis for Ann is that she will certainly die within 21 years, but we do not know exactly when. The risk of dying is 99% before her 50[th] birthday, but there is also a 1% chance that she will die in her 51[st] year. Bob, however, will certainly die before he is 51. For simplicity in this example, although we will use the term "lifetime health", we disregard health/disability adjustment for time with illness and focus only on their age at death. Because there is a 99% probability that Ann will die at a younger age than Bob will, Ann's lifetime health is lower than Bob's in terms of total length of life (past life plus expected future life), even though Bob's expected future life is shorter. This is true even if there is a 1% chance that Ann too will die in her 51[st] year. Health adjustment complicates matters, as we will discuss below, but the principles are the same. Of course, HAAD needs to go beyond hypothetical two-person cases to become a relevant health metric for priority setting in countries with millions of individuals and multiple diseases. HAAD enables comparison of both average lifetime health (aHAAD) and distribution of lifetime health between individuals with different diseases (e.g., at disease onset, Ann's disease will have a very different HAAD distribution than Bob's disease). Methods for calculating aHAAD and the HAAD distribution within disease conditions are presented in the next sections.

### Data

For illustrative purposes, we consider four conditions (acute myeloid leukemia (AML), acute lymphoid leukemia (ALL), schizophrenia, and epilepsy) in six countries (Ethiopia, Haiti, China, Mexico, United States and Japan). The diseases have distinct properties that highlight certain characteristics of HAAD. The two leukemias are fatal, but the incidence of ALL peaks at both young and older age groups, whereas AML incidence peaks at old age groups only. Schizophrenia has large impact on disability over many years, and there are variations in both mortality and morbidity of epilepsy across countries. We consider the leukemias in a US setting, and then we compare schizophrenia and epilepsy across the six countries, representing low-, middle- and high-income settings with different age distributions, levels of health systems development, and access to healthcare among their populations. Estimates for all countries and for all diseases in 2017 are obtained from the freely available GBD results tool [23]. The GBD Study produces cause-specific estimates of deaths, incidence, and prevalence by country, year, age, and sex, utilizing demographic methods [24, 25], ensemble

models [26] using vital registration and verbal autopsy data about causes of death, and Bayesian meta-regression [27] using data from reviews of literature, registries, and hospitals. Table 1 describes variables available in the GBD database, and how they are used to derive other important variables.

**Table 1. Description of data and variables used to calculate Health Adjusted Age at Death (HAAD), the GBD 2017 study [20] is source for all calculations.**

| Variable | Description |
|---|---|
| PIM | *Period of increased mortality*. From expert opinions. Number of years with increased mortality after disease onset. The rate at which mortality declines in the PIM is specific for each condition. For simplicity, PIM = 100 for chronic diseases. We use PIM = 5 for the leukemias. |
| PID | *Period of increased disability*. From expert opinions. Number of years with increased disability after disease onset. The rate at which disability declines in the PID is specific for each condition. For simplicity, PID = 100 for chronic diseases. We use PID = 5 for the leukemias. |
| Pop | *Population size, per 5-year age interval*. From GBD 2017. Transformed to 1-year age intervals by distributing individuals evenly across the five years. |
| $P_D$ | *Prevalence (per capita) of disease per 5-year age interval*. From GBD 2017. Assumed to be the same in all 1-year intervals. |
| $I_D$ | *Incidence (per capita) of disease per 5-year age interval*. From GBD 2017. Assumed to be the same in all 1-year intervals. |
| $M_D$ | *Disease specific rate of death per 5-year age interval, for total population*. From GBD 2017. Assumed to be the same in all 1-year intervals. |
| Q | *All cause probability of death in single-year intervals, for a total population (i.e., baseline mortality)*. From GBD 2017. $q = 1 - \exp(-M_{All\ causes})$ Converted from single-year $M_D$ using common demographic approximation [28]. |
| $YLD_D$ | *Years Lived with Disability (per capita in one year) of disease in 5-year age intervals*. From GBD 2017. Assumed to be the same in all 1-year intervals. |
| **Derived** | |
| $em_D$ | Excess mortality due to disease (case fatality rate). These are not given directly in GBD, but can be calculated using $em_D(age) = \frac{M_D(age)}{P_D(age)}$. This is the extra risk of dying for individuals with disease that is caused directly by the disease itself. Note that this is different from $M_D$, which is the risk of dying from a particular disease for any individual in the population. |
| $q_D$ | Probability of death due to disease and baseline mortality. These are not given directly in GBD, but can be calculated using $q_D(age) = 1 - \exp(-(M_{All\ causes}(age) - M_D(age) + em_D(age)))$. Substituting $em_D$ into $q_D$ yields $q_D(age) = 1 - \exp\left(-\left(M_{All\ causes}(age) + M_D(age)\left(\frac{1}{P_D(age)} - 1\right)\right)\right)$. We can see that if $P_D = 1$, meaning that all individuals in the population have a disease, $q_D$ simply becomes q. This is also the case if there is no mortality from disease, so that $M_D = 0$. |
| $q_D^{PIM}$ | During the period of increased probability of death due to disease, $q_D$ is used for PIM years. After the period, $q_D$ returns to q. |
| Dw | Background disability weight from the population overall (0 is no disability and 1 is death). $dw(age) = YLD_{All\ causes}$. Note that $YLD_{All\ causes}(age)$ is *per capita*. |
| $dw_D$ | Average disability weight due to the disease of interest (0 is no disability and 1 is death). $dw_D(age) = \frac{YLD_D(age)}{P_D(age)}$. Note that $YLD_D(age)$ is *per capita*. |
| $dw_D^{PID}$ | In the GBD study, disability weights are aggregated by subtracting from 1 the product of 1 minus the two disability weights. To calculate the disability weight during the period of increased disability during illness, we can combine the "background" disability (i.e., disability from other causes than D) on average and the disability specifically from the disease on average. $dw_D^{PID}(age) = 1 - (1 - dw_{Background}(age))(1 - dw_D(age))$. To calculate the background disability, we treat YLD rates (i.e., YLD *per capita*) in the population as average disability weights for a given person and solve for the background disability by breaking down the all-cause disability weight into disability from the specific cause and from other causes. $YLD_{All\ Causes}(age) = 1 - (1 - dw_{Background}(age))(1 - YLD_D\ age))$ Solving for the background disability and inserting into the first equation, the disability weight during the period of increased disability is $dw_D^{PID}(age) = 1 - \left(1 - \frac{YLD_{All\ causes}(age) - YLD_D(age)}{1 - YLD_D(age)}\right)(1 - dw_D)$. After the period, dw returns to that of the baseline population (dw). Note that $YLD_{All\ causes}(age)$ and $YLD_D(age)$ are *per capita*. |

The GBD database gives the parameters from Table 1 in 5-year age groups to age 95. The under-5 age group is split into "less than one year old" (<1 group) and "1–4 years old". To obtain single-year age estimates, we undertake the following procedures. For *pop*, we divide the population in the 4- and 5-year age groups evenly by single-year ages, and the terminal age group (95 plus) is divided equally in five parts from 95 to 99. For example, if 500 000 individuals are in the 20–24 age group, we will assume that there are 100,000 20-year olds, 100,000 21-year olds, and so on. For $P_D$, $I_D$, dw, $dw_D$, q(age), and $M_D$(age), we assume that the rates (or disability weights) are the same for each single-year age in the aggregate age groups. For example, if $dw_D$ was 0.2 in the 20–24 age group, we assume that it was 0.2 for 20-year olds, 0.2 for 21-year olds, and so on.

## Part I: Estimating disease-specific age at death

To estimate age at death (AD) from the GBD data described in previous sections, we use standard lifetable methodology, so that AD of an individual is simply

$$AD(age) = age + LY^{future}(age), \tag{1}$$

where "age" is the age of the individual and $LY^{future}$ is the number of life years they have left to live at their respective age. $LY^{future}$ is not known until the individual dies, but its distribution can be estimated as follows. First, we choose Y, the maximum age in our lifetable, by setting the chance of surviving from age Y to Y+1 to zero. In principle, Y can be any age, but we have used Y = 99 throughout this paper. Then we calculate the number of people who are expected to die at different ages in the years to come until they reach age Y. This can be done using an upper diagonal $(Y + 1) \times (Y + 1)$ matrix,

$$\mathbb{N}_{AD} = \begin{bmatrix} N_{0,0} & N_{0,1} & N_{0,2} & \cdots & N_{0,Y} \\ - & N_{1,1} & N_{1,2} & \cdots & N_{1,Y} \\ - & - & N_{2,2} & \cdots & N_{2,Y} \\ - & - & - & \ddots & \vdots \\ - & - & - & \cdots & N_{Y,Y} \end{bmatrix}. \tag{2}$$

In each element of $\mathbb{N}_{AD}$, $N_{c,d}$, the *c* denotes the current age and the *d* denotes the expected age at death. For example, $N_{3,12}$ is the number of today's 3-year-olds who will die at age 12. Each $N_{c,d}$ is calculated using standard lifetable methodology [22], based on the assumption that q(age) remains the same in the future. In other words, $\{N_{c,0}, N_{c,1}, \ldots, N_{c,Y}\}$ is the distribution of expected age at death for an individual with current age *c*. We see that summing the rows, $\sum_{d=0}^{Y} N_{c,d}$, yields the population age structure (pop). Further, summing the columns, $\sum_{c=0}^{Y} N_{c,d}$, gives the number of people that we expect to die at age *d*. That is, the expected number of people for which AD = *d*. The sum $\Sigma_c \Sigma_d N_{c,d}$ is the total current number of people at all ages. In our calculations, we use Y = 99. Note that it follows from the assumption of a static q(age) that average $LY^{future}$ is the same as LE. From (1) we see that AD is dominated by $LY^{future}$ for young children, and by age for very old people.

Before we break the analysis into diseases, we start by analyzing total figures for one country, as this is familiar for most readers and perhaps more intuitive. Fig 1 shows the estimated distribution of AD for all age groups in the total population of the United States in 2017. In the left panel, individuals alive in 2017 are ranked by their current age, and the right panel ranks them by AD. Now we can see that we expect that around 22.6 million people (7%) in the US have AD < 60 years. In comparison, the proportion is much higher in Ethiopia (16%) and

# Age at death for total population in the US

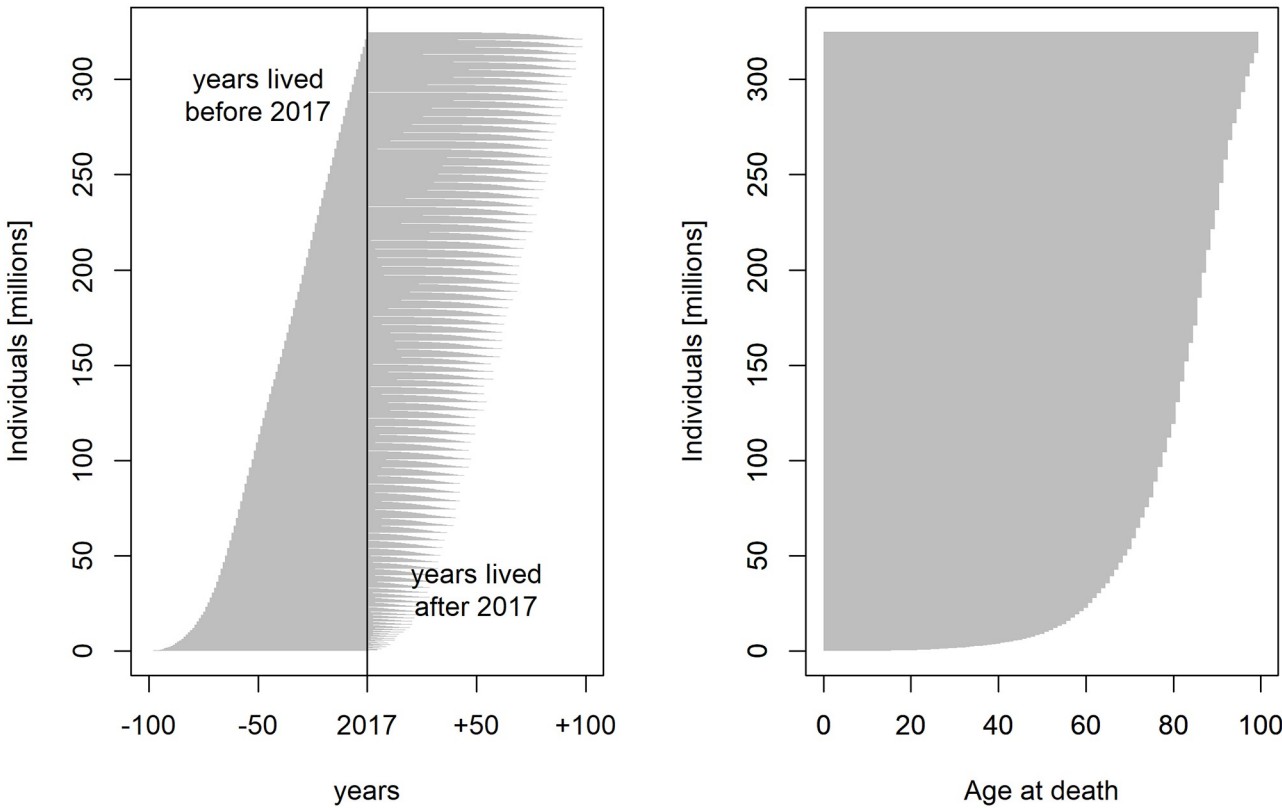

**Fig 1. Estimated distribution of Age at Death (AD) for the total US population (2017).** Left panel: Distribution by age. On the x-axis, -100 corresponds to the year 1917, and +100 corresponds to the year 2117. Years lived before 2017 are observed, whereas years lived after 2017 are expected. Right panel: Distribution by AD.

Haiti (20%). Note that by simply focusing on LE, we would know nothing about such distributional characteristics.

For individuals who get a disease, D, at a particular age, (1) becomes

$$AD_D(age) = age + LY_D^{future}(age). \qquad (3)$$

$LY_D^{future}$ is calculated similarly to $LY^{future}$. Instead of pop(age), we use $I_D(age) \times pop(age)$ (Table 1), and instead of mortality rates for the general population, we use those of individuals with condition D, $q_D$ (Table 1). For diseases with very high mortality, $LY_D^{future}$ will be small for all ages, and $AD_D$ will therefore to a large extent depend on age alone. If the excess mortality is low, the situation resembles that of (1).

Fig 2 shows the estimated AD distribution among the 10,600 people with incident cases of acute myeloid leukemia (AML) and 1,950 people with incident cases of acute lymphoid leukemia (ALL) in the United States in 2017. We see that average AD was 68.8 for AML and 68.5 for ALL. However, as the figure shows, the mean age of onset was 62.5 for AML and 45.6 for ALL, and mean $LY_{AML}^{future}$ was 6.3 years, whereas $LY_{ALL}^{future}$ was 22.9 years. Once again, we see that important information about lifetime health is lost when focusing on averages only.

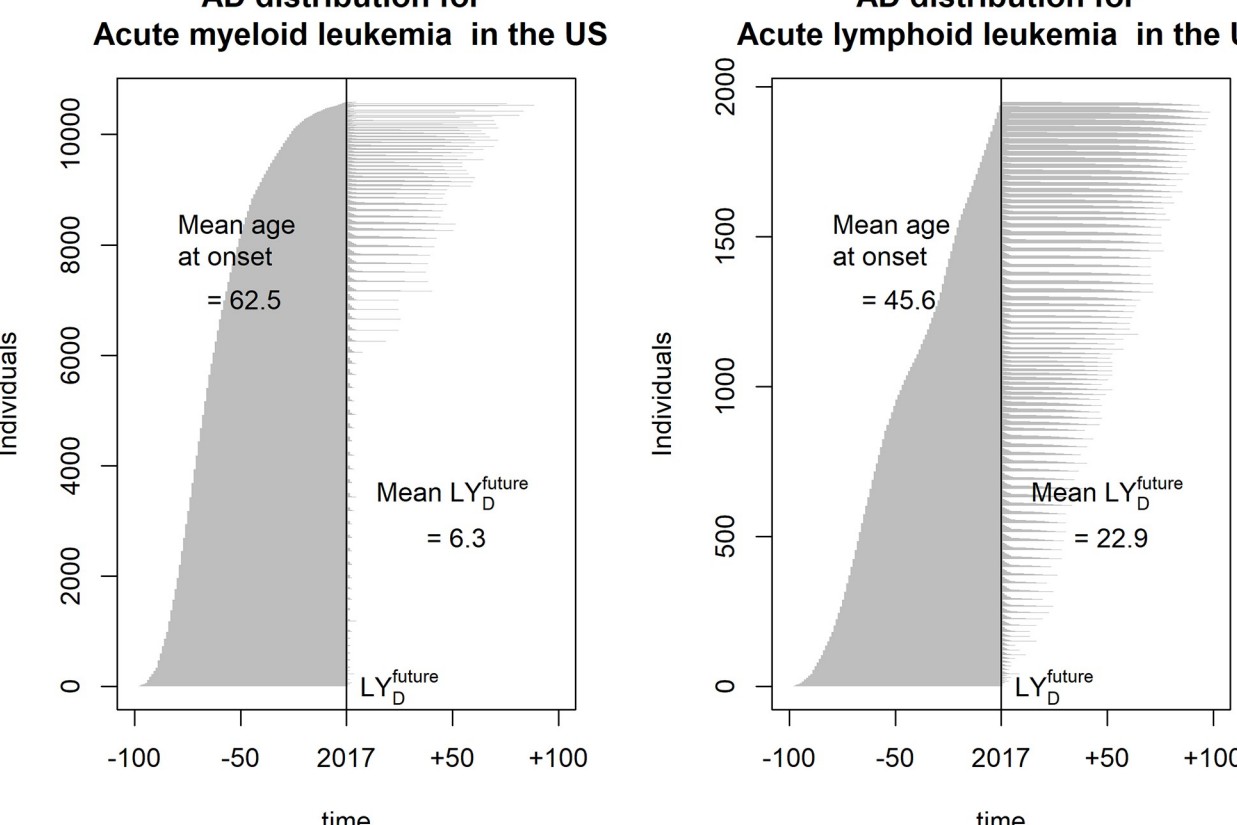

**Fig 2. Estimated Age at Death (AD) for individuals who got leukemia in the US in 2017, sorted according to age at onset.** Left panel: Distribution for acute myeloid leukemia (AML). Right panel: Distribution for acute lymphoid leukemia (ALL).

## Part II. Adjust for morbidity

Non-fatal morbidities should also be considered when assessing lifetime health at disease onset, so that one can compare across fatal and non-fatal diseases with different impacts on health loss. This includes estimating health adjusted age (HAA) and future health adjusted life years (HALY$^{\text{future}}$). Expanding on (3), we get

$$\text{HAAD}_{\text{D}}(\text{age}) = \text{HAA}^{\text{past}}(\text{age}) + \text{HALY}_{\text{D}}^{\text{future}}(\text{age}), \tag{4}$$

In this section we will explain how to estimate HAA$^{\text{past}}$ and HALY$_{\text{D}}^{\text{future}}$ using the baseline disability (i.e., the average disability in the population), dw, and the excess disease-specific disability, dw$_{\text{D}}$ (Table 1).

Fig 3 outlines the conceptual structure of the HAAD method, where both past and future health is summed for everyone with one of the four diseases AML, ALL, epilepsy and schizophrenia.

HAA$^{\text{past}}$ is calculated as follows,

$$\text{HAA}^{\text{past}}(c) = \sum_{i=0}^{c-1}(1 - \text{dw}(i)), \tag{5}$$

where, $c$ is current age and dw(i) is the baseline disability from age "i" to "i+1" (Table 1). We assume that conditions are independent and that past dw are the same regardless of current disease status.

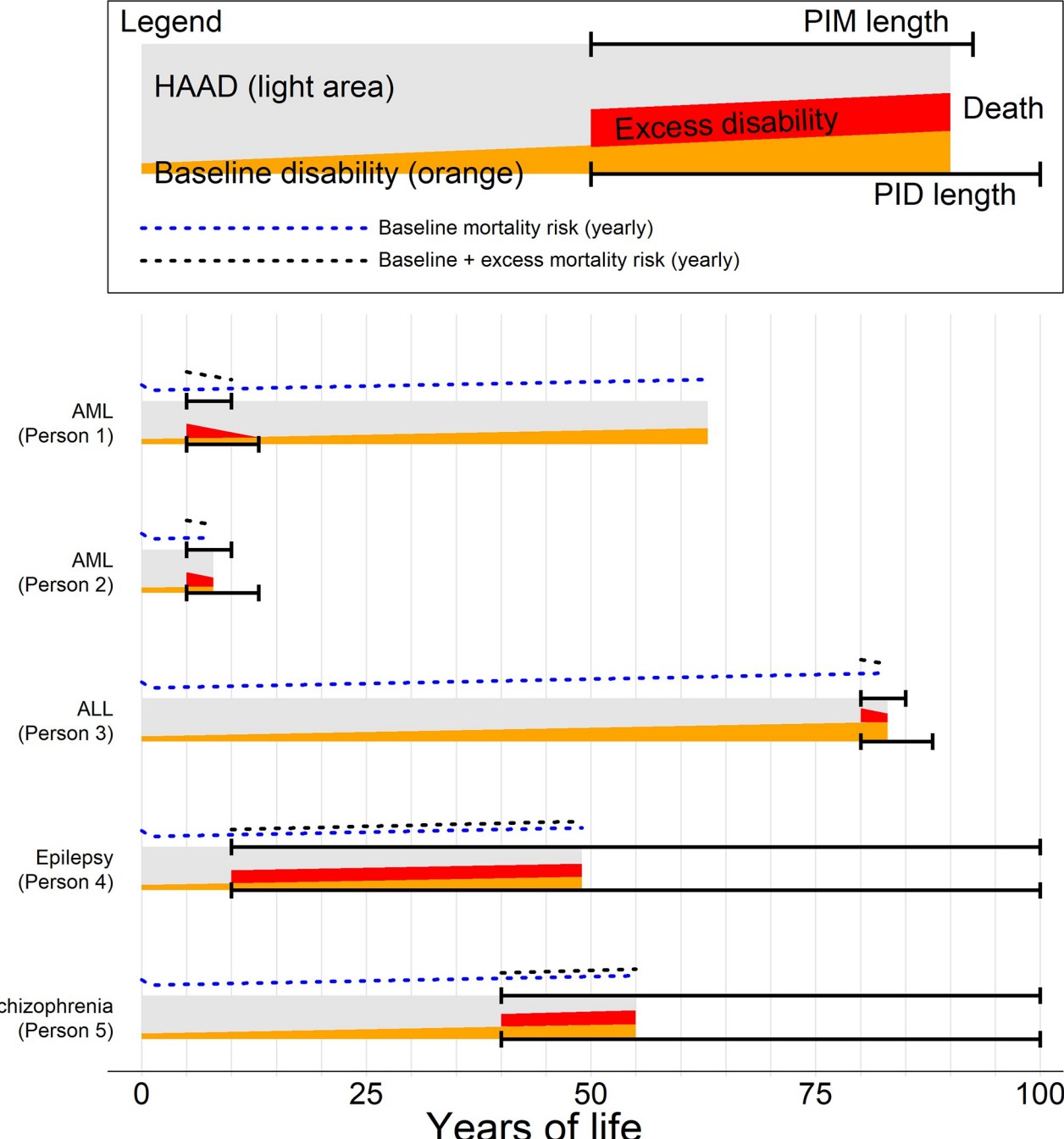

**Fig 3. Outline of the conceptual structure of the Health Adjusted Age at Death (HAAD) method, where we calculate the sum of past health and expected future health for five individuals with different diseases (AML, ALL, epilepsy and schizophrenia).** For a disease D, the dashed black line is baseline mortality (i.e., average mortality in the population), and the blue line is baseline mortality added to the excess risk of death caused by D. The grey area is $HAAD_D$. The orange area is baseline health loss due to disability (dw), and the red area is health loss caused by D ($dw_D$). These areas slope upwards because disability increases with age. The sum of the grey, orange and red areas constitute age at death (AD). The top solid black line gives a period after the onset of D when the person had a period of increased mortality (PIM). The bottom line gives a similar period of increased disability (PID). Note that PIM and PID may be over before death occurs because the individual has survived the course of the disease, as seen in the top example, or PIM and PID may last beyond the death because the person died during the course of the disease, as seen in the other examples.

To account for future non-fatal health loss caused by a disease, D, we use disease specific excess disability, $dw_D$, and mortality, $q_D$, as calculated in Table 1. However, mortality risk is returned from $q_D$ to $q$ after a period of increased mortality (PIM), and morbidity returns from $dw_D$ to $dw$ after a period of increased disability (PID) (Table 1 expands on PIM and PID).

In Fig 3 we see from the AML examples how two different persons may fare under the same PIM and PID. Person 1 survives long enough that both mortality and morbidity return to those of the baseline population, and then dies at age 63 from a different cause, whereas Person 2 dies during the PIM.

Future health adjusted life years, as a function of current age and expected age at death, is

$$\text{HALY}_D^{\text{future}}(c, d) = \begin{cases} 0.5 \times (1 - dw_D^{\text{PID}}(d)), c = d \\ (c + 0.5) - (0.5 \times dw_D^{\text{PID}}(d) + \sum_{i=c}^{d-1} dw_D^{\text{PID}}(i)), c < d, \\ -, c > d \end{cases} \quad (6)$$

where $c$ is current age and $d$ is expected age at death. Note that it is only necessary to sum over time when $c < d$. As in (5), $dw_D^{\text{PID}}(i)$ is the disability weight from age "i" to "i+1". However, because we estimate future health loss, the disability weight must be adjusted during PID. This means that the disability increased for a period of PID years after onset before returning to that of the general population (Table 1).

We next set out to estimate the HAAD distribution in a population. This is done in several steps. First, we create one matrix for past health, and one for future health. The matrix for past health is

$$\mathbb{H}^{\text{past}} = \begin{bmatrix} 0 & 0 & \cdots & \cdots & 0 \\ - & \text{HAA}^{\text{past}}(1) & \cdots & \cdots & \text{HAA}^{\text{past}}(1) \\ - & - & \text{HAA}^{\text{past}}(2) & \cdots & \text{HAA}^{\text{past}}(2) \\ - & - & - & \ddots & \vdots \\ - & - & - & - & \text{HAA}^{\text{past}}(Y) \end{bmatrix}, \quad (7)$$

where $\text{HAA}^{\text{past}}$ is from (5). Because the row number represents current age, we see that the elements are the same within each row. In other words, your past health only depends on your current age, and not your future expected age at death (column number).

In the matrix for future health, we need to account for both current age and expected age at death. Using $\text{HALY}_D^{\text{future}}$ from (6), we get

$$\mathbb{H}_D^{\text{future}} = \begin{bmatrix} \text{HALY}_D^{\text{future}}(0, 0) & \text{HALY}_D^{\text{future}}(0, 1) & \cdots & \text{HALY}_D^{\text{future}}(0, Y) \\ - & \text{HALY}_D^{\text{future}}(1, 1) & \cdots & \text{HALY}_D^{\text{future}}(1, Y) \\ - & - & \ddots & \vdots \\ - & - & - & \text{HALY}_D^{\text{future}}(Y, Y) \end{bmatrix}. \quad (8)$$

Adding past and future health yields

$$\mathbb{H}_D^{\text{HAAD}} = \mathbb{H}^{\text{past}} + \mathbb{H}_D^{\text{future}}. \quad (9)$$

In $\mathbb{H}_D^{\text{HAAD}}$, row number $c$ estimates HAAD for individuals who are $c$ years old, whereas column number $d$ estimates HAAD for individuals who will die at age $d$. As opposed to the discrete AD, HAAD is continuous. For example, one individual who dies at age 80 may have HAAD = 67.3, whereas another could have achieved 67.4 or 67.5. Because we do not have

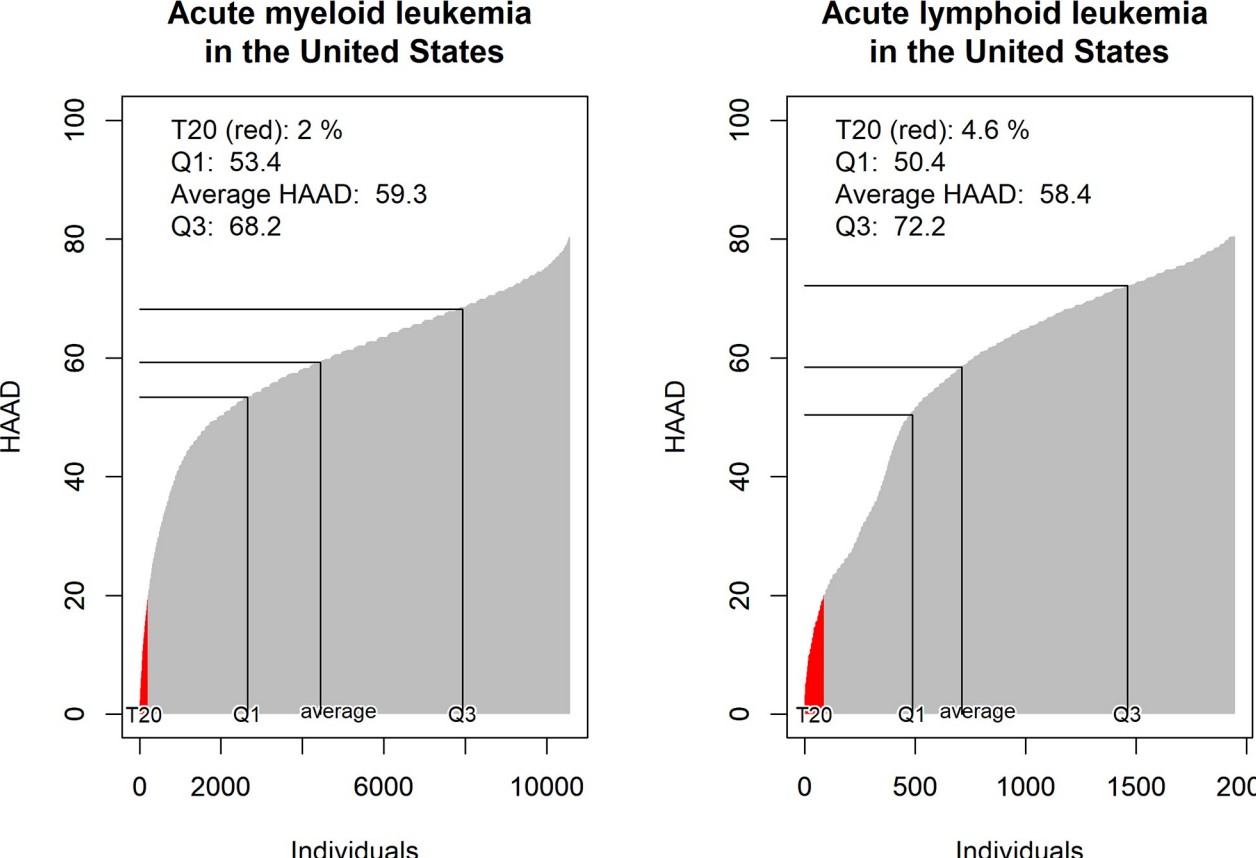

**Fig 4. Estimated distribution of Health Adjusted Age at Death (HAAD) in the United States in 2017.** Left: Acute myeloid leukemia (AML). Right: Acute lymphoid leukemia (AML).

access to data on individuals, every individual with the same condition and the same age of onset is assumed to have the same HAAD distribution. In $\mathbb{N}_{AD}$ from (2), we obtained these distributions by considering the rows. This information is not available in $\mathbb{H}_D^{HAAD}$, but we may create a new matrix, $\mathbb{N}_{AD}^{PIM}$, where the elements correspond to those of $\mathbb{N}_{AD}$, but are calculated using $q_D^{PIM}$ instead of q (Table 1).

Pairing all elements in $\mathbb{N}_{AD}^{PIM}$ with the corresponding element in $\mathbb{H}_D^{HAAD}$ yields the estimated HAAD distribution for all values of *c* and *d* for the disease D.

In $\mathbb{N}_{AD}^{PIM}$ we use incidence to identify those who get the disease D each year, which is especially useful for life-long conditions. See A1 Fig in (S1 Appendix) for details on incidence assumptions that are being used in HAAD calculations for AML and ALL in the US.

Fig 4 shows that even though the estimated aHAAD is similar for AML (59.3) and ALL (58.4), the distribution across individuals is different. For example, in the US we expect 4.6% of individuals with ALL to have HAAD < 20 (T20), compared with only 2.0% for individuals with AML. Hence, considering risk of attaining little lifetime health, individuals with ALL would be worse off. Still, the 75[th] percentile (Q3) of HAAD for individuals with ALL (72.2) was higher than in people with AML (68.2), so with respect to chance of attaining much lifetime health individuals with AML would be the worse off. Again, this highlights the need for distributional concerns in policy making.

**Table 2. Average Health Adjusted Age at Death (aHAAD), T20, Q1 and Q3 for four conditions in six countries, results from 200 NCDI conditions can be found in the Appendix Table A4 (the GBD 2017 study [20] is source for all calculations).**

| | | Acute lymphoid leukemia | Acute myeloid leukemia | Epilepsy | Schizophrenia |
|---|---|---|---|---|---|
| Ethiopia | aHAAD | 34.8 | 32.7 | 36.9 | 41.2 |
| | T20 (%) | 44.8 | 38.7 | 14.0 | 0.5 |
| | Q1 | 8.6 | 8.7 | 26.3 | 35.7 |
| | Q3 | 62.2 | 54.4 | 46.4 | 46.0 |
| Haiti | aHAAD | 35.1 | 35.1 | 39.7 | 40.8 |
| | T20 (%) | 41.8 | 30.1 | 8.2 | 0.5 |
| | Q1 | 9.7 | 14.9 | 30.6 | 35.3 |
| | Q3 | 60.2 | 52.8 | 48.6 | 45.6 |
| China | aHAAD | 58.3 | 48.0 | 56.8 | 45.7 |
| | T20 (%) | 5.0 | 14.3 | 1.8 | 0.1 |
| | Q1 | 48.4 | 33.2 | 50.6 | 39.3 |
| | Q3 | 72.3 | 64.3 | 65.7 | 51.5 |
| Mexico | aHAAD | 46.0 | 41.1 | 53.6 | 44.5 |
| | T20 (%) | 22.3 | 21.5 | 2.0 | 0.2 |
| | Q1 | 21.4 | 22.6 | 46.8 | 38.7 |
| | Q3 | 68.3 | 59.1 | 61.8 | 49.8 |
| US | aHAAD | 58.4 | 59.3 | 58.9 | 43.6 |
| | T20 (%) | 4.6 | 2.0 | 0.7 | 0.2 |
| | Q1 | 50.4 | 53.4 | 53.4 | 38.3 |
| | Q3 | 72.2 | 68.2 | 66.7 | 48.0 |
| Japan | aHAAD | 66.0 | 63.5 | 64.4 | 48.7 |
| | T20 (%) | 1.5 | 1.5 | 0.6 | 0.0 |
| | Q1 | 61.4 | 58.1 | 59.2 | 42.3 |
| | Q3 | 77.2 | 72.9 | 72.5 | 54.5 |

T20: Proportion of individuals with disease who attain HAAD < 20.

Q1: Attained HAAD for the individual at the 25th percentile

Q3: Attained HAAD for the individual at the 75th percentile

## Results

Table 2 shows estimated aHAAD, T20, Q1 and Q3 for ALL, AML, schizophrenia and epilepsy in six countries (see A4 Table in S1 Appendix for 200 NCDI conditions). Rank orders of the four conditions varied both between countries and according to measure of who is worse off.

The estimated HAAD distribution for schizophrenia was similar across the six settings, although Japan stands out in a positive manner. The difference in Q1 for schizophrenia between the US and Japan was larger than the difference between the US and Ethiopia (Haiti 35.3, Ethiopia 35.7, China 39.3, Mexico 38.7, US 38.3, Japan 42.3). The same applies to aHAAD. T20, the proportion of people with an estimated HAAD <20, was low across countries for schizophrenia, which is reasonable, as schizophrenia rarely manifests in childhood. Variability in Q1 for epilepsy between countries was high (Ethiopia 26.3, Haiti 30.6, Mexico 46.8, China 50.6, US 53.4, Japan 59.2), and the estimated HAAD distribution for epilepsy was much more unequal within countries with a low Q1 (difference between quartiles (Q3-Q1) was 19.9 in Ethiopia, 18.0 in Haiti, 15.1 in China, 15.0 in Mexico, 13.3 in the US, and 13.3 in Japan). Further, only 0.6% of the Japanese had an estimated HAAD below 20, but the number was 14.0% among Ethiopians and 8.2% among Haitians.

## Discussion

According to fairness concerns, limited health care resources should be allocated to interventions that benefit the worse off in society [5–7, 18]. In this paper, we present a quantitative method for identifying the worse off by estimating the distribution of lifetime health across individuals in the same disease category. We show how two conditions, ALL and AML, with similar estimated aHAADs have substantially different HAAD distributions. In addition, we show how HAAD varies across countries, and demonstrate how the HAAD distribution captures different aspects of the fact that diseases are typically more severe in low-income than in high-income countries. Our new framework is important for priority setting because it can be used to assign extra value to health gains from interventions targeting the worse off. The relevance of HAAD is particularly good for preventive interventions for a disease where you are likely to capture benefits across a range of ages (for example treating strep throat in school children to prevent rheumatic heart disease).

Sullivan, in 1971, suggested how morbidity adjustment could be done for LE to get HALE by modifying the standard life table model to estimate the expected duration of a condition by exposing a birth cohort of a disease specific mortality and disability rate over a lifetime [22]. Sullivan's method estimates average expected years of healthy life rather than the distribution of HAAD between individuals as done in this paper.

In this article we present HAAD as an achievement measure, but it may be more intuitive to measure shortfall of lifetime health from what someone could potentially achieve. HAAD could be converted to such a gap-measure by using the YLL method applied in GBD. Shortfall in life years could be calculated at disease onset by using the lowest mortality by age in the world as a reference. Shortfall in disabilities could use the lowest YLD rates (i.e., YLD *per capita*) across countries as a reference for disability shortfall. However, disease shortfall measures are beyond the scope of this paper.

PIM and PID, as presented in this paper, have some limitations. They could be different for the same condition across settings, as would be the case for conditions, like HIV, that can be treated or controlled more effectively in some countries than in others. Part of these differences should be captured in the excess mortality differences between countries in our current analysis, but the durations of the periods are also likely to vary. Additionally, PIM and PID do not capture the nature of conditions where mortality and morbidity have complicated temporal patterns. For example, the peak increase in mortality risk for HIV patients is about a decade after onset. At a conceptual level, these obstacles are easy to handle. One simply must estimate PIM and PID for all conditions under consideration in all relevant settings. However, the empirical task of getting precise PIM and PID estimates is not trivial.

Understanding the underlying reasons for differences in HAAD distributions can have policy implications. Observed differences in the distribution between countries can originate from several reasons. It is important to note that as a measure of lifetime health among people with a specific condition, HAAD is influenced by mortality risk and morbidity from other causes, as well as by the age at which the disease occurs. Thus, variation in HAAD could be caused from differences in demography and epidemiology at the country level, or from variations in access to health care that underlie differences in disease-specific morbidity or mortality rates. The relative contribution of each of these differences to the overall difference in HAAD between countries depends on which countries are being compared. How to quantify the role of each factor is discussed further in S1 Appendix of (A2 Fig).

To measure the true disease-specific lifetime health distribution, the past health would be calculated using observed past disability for individuals. The estimates available from the GBD Study used as input for our calculations were limited in several ways. Although estimates for

282 causes of death and 354 conditions from the GBD were available by five-year age groups and both sexes for 195 countries, these estimates are based on sparse data in many low- and middle-income countries and rely heavily on modeled relationships with covariates and other data from the same region. There was no individual-level morbidity and mortality information, so we used population averages. This meant that we were unable to account for correlation between illnesses. For example, people who die of a car accident at age 45 may be different on average from those who die of a myocardial infarction at age 45 regarding lifestyle (smoking, exercise, diet) and biology (metabolism, genetics), which could affect the risk of other morbidities. As a result, our estimates of HAAD may be high (overestimate lifetime health) for illnesses that are often experienced with comorbidities because they do not capture the higher burden from the associated illnesses. Conversely, our estimates of HAAD may be low (underestimate lifetime health) for illnesses that have few comorbidities. These limitations are especially evident in mental health conditions. The GBD estimates do not attribute any mortality to mental health disorders; however, we know that patients with mental health disorders have higher mortality risk compared to the general population [29–31]. Our schizophrenia HAAD results are additionally limited by distributions of disability weights that do not vary along with treatment availability across countries [32]. The limited time series available from the GBD meant that we did not have complete historical average disability rates. For consistency, we used age-specific rates of disability for the calculations of past health; however, health achievement in a real population would use historical disability information if available.

## Conclusion

Increasing availability of demographic and epidemiologic data creates opportunities for estimating the lifetime health at disease onset to guide priority setting in health care. Policy makers, supported by the ethical literature, may want to give higher priority to the worse off. However, the impact of such fairness concerns to health policy does not match the impact cost-effectiveness analysis has had on policy the last decades. Data availability and lack of rigorous methods for estimating the lifetime health at time of disease onset are likely contributing factors to the negligence of the worse off in de facto health care priority setting. Here we have presented a method for estimating lifetime health by considering HAAD, illustrated with examples how to estimate the distribution of HAAD across individuals, and shown why considering these distributions is relevant for priority setting in health care and the measurement of population health. We lay the foundations for undertaking detailed calculations of disease-specific HAAD in multiple countries.

## Supporting information

**S1 Appendix. [20], [33].**
(DOCX)

## Author Contributions

**Conceptualization:** Kjell Arne Johansson, Jan-Magnus Økland, Ole Frithjof Norheim, Matthew M. Coates, Øystein Ariansen Haaland.

**Data curation:** Kjell Arne Johansson, Jan-Magnus Økland, Eirin Krüger Skaftun.

**Formal analysis:** Kjell Arne Johansson, Jan-Magnus Økland, Eirin Krüger Skaftun, Matthew M. Coates, Øystein Ariansen Haaland.

**Funding acquisition:** Kjell Arne Johansson, Gene Bukhman, Ole Frithjof Norheim.

**Investigation:** Kjell Arne Johansson, Jan-Magnus Økland, Ole Frithjof Norheim, Matthew M. Coates, Øystein Ariansen Haaland.

**Methodology:** Kjell Arne Johansson, Jan-Magnus Økland, Eirin Krüger Skaftun, Gene Bukhman, Ole Frithjof Norheim, Matthew M. Coates, Øystein Ariansen Haaland.

**Project administration:** Kjell Arne Johansson, Ole Frithjof Norheim, Øystein Ariansen Haaland.

**Software:** Jan-Magnus Økland, Matthew M. Coates, Øystein Ariansen Haaland.

**Supervision:** Kjell Arne Johansson, Gene Bukhman, Ole Frithjof Norheim, Øystein Ariansen Haaland.

**Visualization:** Kjell Arne Johansson, Jan-Magnus Økland, Matthew M. Coates, Øystein Ariansen Haaland.

**Writing – original draft:** Kjell Arne Johansson.

**Writing – review & editing:** Kjell Arne Johansson, Jan-Magnus Økland, Eirin Krüger Skaftun, Gene Bukhman, Ole Frithjof Norheim, Matthew M. Coates, Øystein Ariansen Haaland.

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
