## [Decision Letter · Decision Letter 0]

10 Oct 2019

PONE-D-19-18816

Measuring Baseline Health with Individual Health-Adjusted Life Expectancy (iHALE)

PLOS ONE

Dear Mr. Haaland,

Thank you for submitting your manuscript to PLOS ONE. After careful consideration, we feel that it has merit but does not fully meet PLOS ONE’s publication criteria as it currently stands. Therefore, we invite you to submit a revised version of the manuscript that addresses the points raised during the review process.

We would appreciate receiving your revised manuscript by Nov 24 2019 11:59PM. To enhance the reproducibility of your results, we recommend that if applicable you deposit your laboratory protocols in protocols.io, where a protocol can be assigned its own identifier (DOI) such that it can be cited independently in the future. For instructions see: http://journals.plos.org/plosone/s/submission-guidelines#loc-laboratory-protocols

A **rebuttal letter** that responds to **EACH** point raised by the academic editor and reviewer(s). This letter should be uploaded as separate file and labeled 'Response to Reviewers'.A **marked-up copy** of your manuscript that highlights changes made to the original version. This file should be uploaded as separate file and labeled 'Revised Manuscript with Track Changes'.An **unmarked version** of your revised paper without tracked changes. This file should be uploaded as separate file and labeled 'Manuscript'.

We look forward to receiving your revised manuscript.

Kind regards,

Brecht Devleesschauwer

Academic Editor

PLOS ONE

Journal Requirements:

2. Please include additional information on how experts' opinions (which were used to determine some of the variables described in table 1) were collected and analysed.

Additional Editor Comments (if provided):

Reviewers' comments:

Reviewer's Responses to Questions

**Comments to the Author**

1. Is the manuscript technically sound, and do the data support the conclusions?

Reviewer #1: Yes

Reviewer #2: Yes

2. Has the statistical analysis been performed appropriately and rigorously? 

Reviewer #1: N/A

Reviewer #2: Yes

3. Have the authors made all data underlying the findings in their manuscript fully available?

Reviewer #1: Yes

Reviewer #2: Yes

4. Is the manuscript presented in an intelligible fashion and written in standard English?

Reviewer #1: Yes

Reviewer #2: Yes

5. Review Comments to the Author

Reviewer #1: The authors present an interesting variant of the HALE measure aimed to help with considerations of health equity for policy decisions making, and show how it can capture variation across diseases/countries that other health expectancy/gap measures cannot. One weakness of the measure has to do with co-morbidity, which the authors adequately discuss.

Major comments

1. I am uneasy with the term ‘baseline health’ used to define both past and expected future health. ‘Baseline’ makes me think of the health status of an individual/population at a point in time, rather than also being ‘conditioned’ on future health. Is there another, better term available?

2. As iHALE is a distributional measure, reporting average iHALE does not seem insightful (except for when pointing out that two conditions can have similar average iHALE but differ when looking at the distribution, as the authors do). It would make sense to background the average measure in the text.

3. More information on GBD sources are required, as characteristics - particularly the reliability of age-specific prevalence – would appear vital to the computations of iHALE. Are the illness data adjusted or modelled in any way, or are they taken directly from surveillance systems or routine recording? For some of the countries I am sceptical that data by 5-year age-group would have been available. Please consider the limitations of the GBD source.

Other comments

1. There are no page numbers, so I will have to point at sections/line numbers. Intro lines 19-20: the focus is described as how a particular illness affects total lifetime health. It is only future health that can be affected; please edit.

2. I wonder about the assertion that iHALE ‘measures … health at an individual level’ (Abstract and elsewhere); in theory perhaps, but in practice (as demonstrated in the current paper), it is computed for strata of all individuals with a certain condition at a certain age. The authors may wish to re-think the use of ‘individual ’ and edit appropriately.

3. The description of YLD as ‘in one particular year’ and ‘during the year’ is only relevant for the prevalence-based approach to computing YLD and DALY, such as used by the GBD project; please clarify.

4. HALE needs to be defined at first mention.

5. Methods line 8. If Ann can die in her 51st year, then it seems that she ‘will most certainly die within 21 years’ (not 20).

6. Methods/Data lines 9-10. Edit needed - for disease prevalence, the source must not only be the GBD ‘cause of death’ database.

7. Concl line 4: ‘concerned about giving higher priority’ is unfortunately ambiguous (ie. want to.. or want to avoid..)

8. Concl line 10. ‘measure that is sensitive to distribution’. This is a bit awkward.. ‘distribution of what’? Please reword, given that iHALE is a distribution, and summary statistics (T20, quartiles etc) can be produced from it.

9. Fig 2 : the word ‘age’ in the plot area is distracting and not described. Also suggest to change ‘Mean age’ to ‘Mean age of onset’

10. Fig 3 is very hard to interpret. In the top panel, does ‘Death’ refer to the red area or to a point in time (on x-axis)? Should the rightmost endcap on PID length not be flush with the right border of the orange area? The text in the red area is illegible. Why do coloured areas slope upwards? Please improve and edit accompanying description.

Reviewer #2: This paper calculates health adjusted life expectancy for patient groups, and separates this in an already realised part (“past”) and still to realise part. This is an interesting way to look at differences between groups, and yields a different perspective from other summary measures of public health. Therefore I think this warrants publication, although I am not yet completely convinced of its importance of health policy making. But it surely adds something to information from other measures. Furthermore, I do not agree that these are outcomes on the level of the individual, as these are measures for groups: groups with the same age and disease. So the i in iHALE to me is not justified.

Page 13 is confusing, as it is unclear what are the data that are used here. Only half way down the page I understood that this are data constructed from life table data, and this part is explaining the conceptual calculations behind the measure. An example of something that put me on the wrong foot was: “First, we define Y, the maximum age in our lifetable, by setting the chance of surviving from age Y to Y+1 to zero.” This is not a definition of Y, but a statement that one arbitrarily is assuming a particular Y. Reading this I also want to understand what value Y is used, which is only mentioned much later.

Equation (7) needs more explanation, as I can not grasp what is happening here; I seem to me that there is some summation over time missing.

In table 1 it is not clear whether the event rates are rates or one-year probabilities.

Also in table 1 it is unclear to me why dw is divided by 100000. YLD is normally given for a population. Is YLD in the formula the YLD for a population of 100000? Is this age dependent? More explanation is needed, also for the formula used in the last row of the table.

6. PLOS authors have the option to publish the peer review history of their article (what does this mean?). If published, this will include your full peer review and any attached files.

Reviewer #1: No

Reviewer #2: No

---

## [Author Response · Author response to Decision Letter 0]

8 Jan 2020

Review Comments to the Author

Reviewer #1: 

Major comments

1. I am uneasy with the term ‘baseline health’ used to define both past and expected future health. ‘Baseline’ makes me think of the health status of an individual/population at a point in time, rather than also being ‘conditioned’ on future health. Is there another, better term available?

Reply: We agree that this term is confusing, and now use “health status” throughout the paper. 

2. As iHALE is a distributional measure, reporting average iHALE does not seem insightful (except for when pointing out that two conditions can have similar average iHALE but differ when looking at the distribution, as the authors do). It would make sense to background the average measure in the text.

Reply: This is a good point. We have changed the text (Abstract, last sentence of Results) to more frequently use metrics other than the average:

“Many illnesses, such as epilepsy, are associated with a higher health status in high-income settings (Q1 in Japan=59.1) than in low-income settings (Q1 in Ethiopia=26.3).” 

Page 15, paragraph 2:

“The difference in Q1 for schizophrenia between the US and Japan was larger than the difference between the US and Ethiopia (Haiti 35.3, Ethiopia 35.6, China 39.3, Mexico 38.7, US 38.2, Japan 42.1). The same applies to the average HEAD.”

Also page 15, paragraph 2: 

“Variability in Q1 for epilepsy between countries was high (Ethiopia 26.3, Haiti 30.5, Mexico 46.8, China 50.4, US 53.3, Japan 59.1). In addition, there is a much more unequal HEAD distribution for epilepsy in countries with a low Q1 (difference between quartiles (Q3-Q1) was 19.9 in Ethiopia, 17.9 in Haiti, 15.2 in China, 14.9 in Mexico, 13.3 in Japan, and 13.2 in the US).”

3. More information on GBD sources are required, as characteristics - particularly the reliability of age-specific prevalence – would appear vital to the computations of iHALE. Are the illness data adjusted or modelled in any way, or are they taken directly from surveillance systems or routine recording? For some of the countries I am sceptical that data by 5-year age-group would have been available. Please consider the limitations of the GBD source.

Reply: We agree that more information could be added to the paper. We now write (page 8, paragraph 1):

“Estimates for all countries and for all diseases in 2017 are obtained from the freely available GBD results tool [23]. The GBD Study produces cause-specific estimates of deaths, incidence, and prevalence by country, year, age, and sex, utilizing demographic methods [24, 25], ensemble models [26] using vital registration and verbal autopsy data about causes of death, and Bayesian meta-regression [27] using data from reviews of literature, registries, and hospitals.”

and we discuss several limitations of the input data on page 18, paragraph 1. We have added to that paragraph.

“Although estimates for 282 causes of death and 354 conditions from the GBD were available by five-year age groups and both sexes for 195 countries, these are based on sparse data in many low- and middle-income countries and rely heavily on modeled relationships with covariates and other data from the same region.” 

Other comments

1. Intro lines 19-20: the focus is described as how a particular illness affects total lifetime health. It is only future health that can be affected; please edit.

Reply: We agree that the illness itself only affects future expected health. Differences in the distribution of age of onset of diseases also mean that past health differs across diseases. We have more explicitly described these two components of lifetime health in this section. The text now reads (second paragraph of Introduction, final sentence): 

“Technically, only the future health (after disease onset) is affected directly by the disease, but lifetime health is the sum of past (before disease onset) and future health expectancies. These differences in years of healthy life lived before disease onset among people with different diseases also inform the potential lifetime health that can be attained and are part of measuring health status by the view we use.”

2. I wonder about the assertion that iHALE ‘measures … health at an individual level’ (Abstract and elsewhere); in theory perhaps, but in practice (as demonstrated in the current paper), it is computed for strata of all individuals with a certain condition at a certain age. The authors may wish to re-think the use of ‘individual ’ and edit appropriately.

Reply: We agree, and now use health expected age at death (HEAD) instead of iHALE. Further, the text has been changed as follows (Abstract, Methods, first sentence): 

“HEAD measures health status for individuals with specific conditions, ...”

and (Methods, first sentence)

“HEAD measures lifetime health for individuals with specific conditions, …”

and (Discussion, third paragraph, 2nd and 3rd sentences)

“HEAD could be converted to a gap-measure by using the YLL method applied in GBD. Shortfall in life years could be calculated at disease onset by using the lowest mortality by age in the world as a reference.” 

3. The description of YLD as ‘in one particular year’ and ‘during the year’ is only relevant for the prevalence-based approach to computing YLD and DALY, such as used by the GBD project; please clarify. 

Reply: We have specified this in the text. (Introduction, 4th paragraph, 3rd sentence: 

“GBD uses a prevalence-based approach, where DALYs are calculated for a set of diseases by summing the years of life lost (YLL) compared to a reference life expectancy and years lived with disability (YLD) in one particular year due to each disease”

4. HALE needs to be defined at first mention.

Reply: The HALE abbreviation is defined in the fourth paragraph in the Introduction. We have added another sentence to this paragraph that describes HALE:

“HALE measures the life expectancy in a population, adjusting for the disability experienced in the population, using age-specific mortality rates and YLDs per capita [22].”

5. Methods line 8. If Ann can die in her 51st year, then it seems that she ‘will most certainly die within 21 years’ (not 20).

Reply: We agree, and this has been corrected accordingly

6. Methods/Data lines 9-10. Edit needed - for disease prevalence, the source must not only be the GBD ‘cause of death’ database.

Reply: We have corrected this mistake, and now refer to “GBD results tool” (second to last sentence, first paragraph in Data). We also changed the reference to the GBD results tool website. 

7. Concl line 4: ‘concerned about giving higher priority’ is unfortunately ambiguous (ie. want to.. or want to avoid..)

Reply: We agree, and have rephrased (first paragraph in Conclusion, second sentence): “Policy makers, supported by the ethical literature, may want to give higher priority to the worse off.” 

8. Concl line 10. ‘measure that is sensitive to distribution’. This is a bit awkward.. ‘distribution of what’? Please reword, given that iHALE is a distribution, and summary statistics (T20, quartiles etc) can be produced from it.

Reply: We agree that this was vague, and the text now says (second to last sentence in Conclusions): 

“Here we have presented a method for calculating HEAD, illustrated with examples how to estimate the distribution of health across individuals, and shown why considering these distributions is relevant for priority setting in health care and the measurement of population health.”

9. Fig 2 : the word ‘age’ in the plot area is distracting and not described. Also suggest to change ‘Mean age’ to ‘Mean age of onset’

Reply: We have removed “age” from the figure, and corrected “Mean age” as the reviewer suggested. 

10. Fig 3 is very hard to interpret. In the top panel, does ‘Death’ refer to the red area or to a point in time (on x-axis)? Should the rightmost endcap on PID length not be flush with the right border of the orange area? The text in the red area is illegible. Why do coloured areas slope upwards? Please improve and edit accompanying description.

Reply: The figure text now reads (changes highlighted in *...*):

“Outline of the conceptual structure of the *Health Expected Age at Death (HEAD)* method, where we calculate the sum of past health and expected future health for five individuals with different diseases (AML, ALL, epilepsy and schizophrenia). For a disease D, the dashed black line is background mortality, and the blue line is background mortality added to the excess risk of death caused by D. *The grey area is HEADD*. The orange area is background health loss due to disability (dw), and the red area is health loss caused by D (dwD). *These areas slope upwards because disability increases with age.* The sum of the grey, orange and red areas constitute *expected age at death (EAD)*. The top solid black line gives a period after the onset of D when the person had a period of increased mortality (PIM). The bottom line gives a similar period of increased disability (PID). *Note that PIM and PID may be over before death occurs because the individual has survived the course of the disease, as seen in the top example, or PIM and PID may last beyond the death because the person died during the course of the disease, as seen in the other examples.*” 

Reviewer #2: 

I do not agree that these are outcomes on the level of the individual, as these are measures for groups: groups with the same age and disease. So the i in iHALE to me is not justified.

Reply: We agree, and now use HEAD instead of iHALE (see reply to Reviewer 1). 

Page 13 is confusing, as it is unclear what are the data that are used here. Only half way down the page I understood that this are data constructed from life table data, and this part is explaining the conceptual calculations behind the measure. An example of something that put me on the wrong foot was: “First, we define Y, the maximum age in our lifetable, by setting the chance of surviving from age Y to Y+1 to zero.” This is not a definition of Y, but a statement that one arbitrarily is assuming a particular Y. Reading this I also want to understand what value Y is used, which is only mentioned much later.

Reply: We have edited the text as follows (Part I: Estimating disease-specific expected age at death, changes in *...*): 

“*To estimate expected age at death (EAD) from the GBD data described in previous sections, we use standard lifetable methodology, so that EAD* of an individual is simply

 EAD(age)=age+LY^future (age) , (1)

where “age” is the age of the individual and LYfuture is the number of life years they have left to live at their respective age. LYfuture is not known until the individual dies, but its distribution can be estimated as follows. First, we choose Y, the maximum age in our lifetable, by setting the chance of surviving from age Y to Y+1 to zero. *In principle, Y can be any age, but we have used Y=99 throughout this paper.*”

Equation (7) needs more explanation, as I can not grasp what is happening here; I seem to me that there is some summation over time missing.

Reply: We have added the following text to clarify: 

“…where c is current age and d is age at death. Note that it is only necessary to sum over time when c<d.” 

In table 1 it is not clear whether the event rates are rates or one-year probabilities.

Reply: From GBD, prevalence, incidence, and YLDs are rates, not probabilities (the denominator is mid-year population estimates from GBD, which is used as the denominator for rates in GBD). Deaths from GBD were downloaded as rates for 5-year age groups for a single year. To do calculations for specific ages, we assumed the same rate for each single age in the age group. We treated these 1-year rates as probabilities since we’re adding rates to rates. Since we used rates (mx) and treated it as probabilities (qx), there could be small differences in results if anyone tries to replicate using qx from GBD.

Also in table 1 it is unclear to me why dw is divided by 100000. YLD is normally given for a population. Is YLD in the formula the YLD for a population of 100000? Is this age dependent? 

Reply: The initial downloaded value is the rate of YLDs per 100,000. This is also true for prevalence and incidence, though we specified in the Table that those metrics are given per population. We have changed the description to reflect that YLDs are also given per population and changed the equations for dw and dwD accordingly. These rates are age-specific, which we have added to the formula. These are the new formulas:

dw(age)=YLD_(All causes) (age)

dw_D (age)=(YLD_D (age))/(P_D (age))

More explanation is needed, also for the formula used in the last row of the table.

Reply: We have added detail on the last row of the table for the disability weight during the period of increased disability. The description now reads as follows: 

“In the GBD study, disability weights are aggregated by subtracting from 1 the product of 1 minus the two disability weights. To calculate the disability weight during the period of increased disability during illness, we can combine the “background” disability from other causes on average and the disability specifically from the disease on average. 

dw_D^PID (age)=1-(1-dw_Background (age))(1-dw_D (age)) .

To calculate the background disability, we treat YLD rates in the population as average disability weights for a given person and solve for the background disability by breaking down the all-cause disability weight into disability from the specific cause and from other causes.

YLD_(All Causes) (age)=1-(1-dw_Background (age))(1-YLD_D (age))

Solving for the background disability and inserting into the first equation, the disability weight during the period of increased disability is 

dw_D^PID (age)=1-(1-(〖YLD 〗_(All causes) (age)-YLD_D (age) )/(1-YLD_D ))(1-dw_D ) .

After the period, dw returns to that of the background population (dw).”

---

## [Decision Letter · Decision Letter 1]

2 Mar 2020

PONE-D-19-18816R1

Measuring Health Expected Age at Death (HEAD)

PLOS ONE

Dear Mr. Haaland,

Thank you for submitting your manuscript to PLOS ONE. After careful consideration, we feel that it has merit but does not fully meet PLOS ONE’s publication criteria as it currently stands. Therefore, we invite you to submit a revised version of the manuscript that addresses the points raised during the review process.

We would appreciate receiving your revised manuscript by Apr 16 2020 11:59PM. To enhance the reproducibility of your results, we recommend that if applicable you deposit your laboratory protocols in protocols.io, where a protocol can be assigned its own identifier (DOI) such that it can be cited independently in the future. For instructions see: http://journals.plos.org/plosone/s/submission-guidelines#loc-laboratory-protocols

A **rebuttal letter** that responds to **EACH** point raised by the academic editor and reviewer(s). This letter should be uploaded as separate file and labeled 'Response to Reviewers'.A **marked-up copy** of your manuscript that highlights changes made to the original version. This file should be uploaded as separate file and labeled 'Revised Manuscript with Track Changes'.An **unmarked version** of your revised paper without tracked changes. This file should be uploaded as separate file and labeled 'Manuscript'.

We look forward to receiving your revised manuscript.

Kind regards,

Brecht Devleesschauwer

Academic Editor

PLOS ONE

Additional Editor Comments (if provided):

Both reviewers appreciated the revisions made by the authors, but made some further suggestions for improvement. In your revision note, please include EACH comment of the reviewers, provide your reply, and when relevant, include the modified/new text (or motivate why you decided not to modify the text).

Reviewers' comments:

Reviewer's Responses to Questions

**Comments to the Author**

1. If the authors have adequately addressed your comments raised in a previous round of review and you feel that this manuscript is now acceptable for publication, you may indicate that here to bypass the “Comments to the Author” section, enter your conflict of interest statement in the “Confidential to Editor” section, and submit your "Accept" recommendation.

Reviewer #1: (No Response)

Reviewer #2: (No Response)

2. Is the manuscript technically sound, and do the data support the conclusions?

Reviewer #1: Yes

Reviewer #2: Partly

3. Has the statistical analysis been performed appropriately and rigorously? 

Reviewer #1: N/A

Reviewer #2: No

4. Have the authors made all data underlying the findings in their manuscript fully available?

Reviewer #1: No

Reviewer #2: Yes

5. Is the manuscript presented in an intelligible fashion and written in standard English?

Reviewer #1: Yes

Reviewer #2: Yes

6. Review Comments to the Author

Reviewer #1: In this revised version and response letter I believe the authors have adequately addressed all comments and concerns raised on the originally submitted manuscript. However, there is now a overlap of terminology, especially caused by the change of name of the new distributional measure they are proposing - HEAD - and the reader will be as confused as the reviewer.

First, why (besides forming a convenient English-acronym) was the change to HEAD made? 'Health expected age at death' does not make sense to me. 'Expected age at death' does, including the provision (Eq 4) that the expected life years remaining is dependent on the disease that an individual has at age 'age'. The vital difference between EAD and the HEAD measure (if I understand correctly) is that in the latter life-years are adjusted for experienced disability/non perfect health; this is the main reason why 'age at death' is not an accurate descriptor. Please improve the name. Related to this comment, T2 and other reported distribution-summaries refer to 'healthy life years' in the text when I guess they should have referred to 'HEAD; this is confusing.

Second, what is the difference between 'health status' and 'lifetime health'? They seem to be used interchangeably throughout the paper. 'Health status at disease onset' should not include the 'future health' component (but is used - I think - when discussing HEAD, for instance 1st sentence of Conclusion, and in 3rd para of Introduction (it is not intuitive to me that young MS patients have a 'low health status').

Third, what do 'background mortality' and 'background disability' mean? Please explicitly define.

Reviewer #2: The new version is somewhat improved, but important points have not yet been resolved, especially with regards to the formula in table 1. Below my specific comments.

1. The proposed metric now has a new name. Although this omits "individual", it still does not cover the content of the new concept, as the concept seems to be “expected healthy years lived until death”

In this sense I would considered this to be simple a HALE, but calculated for a subgroup of the population (the group that is diagnosed with a particular disease at a certain age). Therefore I wonder if a new term is needed at all. I would suggest a name that reflects this is a type of HALE (e.g. HALE-AD ) (HALE by age of disease incidence).

2. One big problem with this paper is that it seems to use the terms YLD and dw at some places as a kind of synonyms, while these are quit different concepts.The authors should fix this problem with the paper, as this prevents me from fully understanding their calculations.

Some examples of this:

Table 1:

dw(age) = YLDAll causes(age)

How can this be? YLD is a number for the population, which will be much larger then 1, while dw is between 0 and 1.

Because of this I can also no follow the next part in the last row of the table:

YLD.All Causes(age) = 1 − (1 − dwBackground(age))(1 − YLDD(age))

Again, 1 - YLD does not make sense to me, as YLD usually is much larger than 1.

Similarly, in the text of the introduction: ”HALE measures the life expectancy in a population, adjusting for the disability experienced in the population, using age-specific mortality rates and YLDs per capit”. HALE does not use YLDs but dw’s.

3. The calculations treat rates as probabilities. This will work reasonably as long as rates are low. However, they will no longer be so at high ages. Why do you not simple use prob =1- exp(-rate*interval length) (assuming a constant rate in the interval) to calculate the probability?

4. Methods, line 13: The use of “health status” here is confusing, as the example only refers to being alive.

5. Methods, line 21: Here it is claimed that HEAD also looks at distributions, while in the example only the average (=expectancy) is used. I therefore could not follow the reasoning in this part.

6. At parts the text still pertains that the metric is for individuals (while the metric is for groups of individuals). For instance in the conclusion:

“Here we have presented a method for calculating HEAD, illustrated with examples how to estimate the distribution of health across individuals”

7. In the legend of figure 3 it is strange that PID is shorter than PIM. Although this is just a legend, it confuses.

7. PLOS authors have the option to publish the peer review history of their article (what does this mean?). If published, this will include your full peer review and any attached files.

Reviewer #1: No

Reviewer #2: No

---

## [Author Response · Author response to Decision Letter 1]

11 May 2020

Response to reviewers, part II

Dear editor

Please find our replies to the reviewers below. 

Reviewer #1: 

 First, why (besides forming a convenient English-acronym) was the change to HEAD made? 'Health expected age at death' does not make sense to me. 'Expected age at death' does, including the provision (Eq 4) that the expected life years remaining is dependent on the disease that an individual has at age 'age'. The vital difference between EAD and the HEAD measure (if I understand correctly) is that in the latter life-years are adjusted for experienced disability/non perfect health; this is the main reason why 'age at death' is not an accurate descriptor. Please improve the name.

 We have now changed the name to Health Adjusted Age at Death (HAAD), and hope that this addresses the concern of the reviewer. Both “Life Expectancy” and “Age at Death” are measured in years, and these years can be health adjusted. In principle, HALE at birth is the same as expected HAAD at birth. 

We will also get back to the name of the metric in our reply to R2.

 Related to this comment, T2 and other reported distribution-summaries refer to 'healthy life years' in the text when I guess they should have referred to 'HEAD; this is confusing.

 We agree, and now use HAAD where appropriate (twice in the final sentence in Methods section in the Abstract). 

 Second, what is the difference between 'health status' and 'lifetime health'? They seem to be used interchangeably throughout the paper. 'Health status at disease onset' should not include the 'future health' component (but is used - I think - when discussing HEAD, for instance 1st sentence of Conclusion, and in 3rd para of Introduction (it is not intuitive to me that young MS patients have a 'low health status').

 We think it is appropriate to use “health status” early in the Introduction, when we discuss the three perspectives current health, future health, and lifetime health. However, starting from the fourth paragraph of the Introduction, we now use “lifetime health” instead of “health status”. 

 We have changed the last sentence in the third paragraph of the Introduction: “Conditions that are severe from a lifetime health perspective, like multiple sclerosis in young patients, do not necessarily require urgent interventions. To underscore that we use a lifetime health perspective when measuring health status, we will from now on use the term “lifetime health” instead of “health status”.”

 Third, what do 'background mortality' and 'background disability' mean? Please explicitly define.

 We agree that this was confusing, and have made the following changes in the manuscript: 

 We have introduced “baseline mortality” and “baseline disability” to separate between the mortality and disability among all people in the population (baseline) and mortality and disability among all people in the population without condition D (background). 

 The first sentence below Eq. (5) now reads “In this section we will explain how to estimate HAApast and HALY_D^future using the baseline disability (i.e., the average disability in the population), dw, and the excess disease-specific disability, dwD (Table 1).”

 The row explaining q in Table 1 now reads: “All cause probability of death in single-year intervals, for a total population (i.e., baseline mortality).”

 The legend and figure text of Figure 3 now use “baseline” instead of “background”, and the text includes the sentences: 

“For a disease D, the dashed black line is baseline mortality (i.e., average mortality in the population), …”

 We use “background disability” and “background mortality” to describe the disability and mortality of those individuals in the population who do not have condition D. 

 The row explaining dw_D^PID now includes the sentence “To calculate the disability weight during the period of increased disability during illness, we can combine the “background” disability (i.e., disability from other causes than D) on average and the disability specifically from the disease on average.”

Reviewer #2: 

 The proposed metric now has a new name. Although this omits "individual", it still does not cover the content of the new concept, as the concept seems to be “expected healthy years lived until death”

In this sense I would considered this to be simple a HALE, but calculated for a subgroup of the population (the group that is diagnosed with a particular disease at a certain age). Therefore I wonder if a new term is needed at all. I would suggest a name that reflects this is a type of HALE (e.g. HALE-AD ) (HALE by age of disease incidence).

 We have now changed the name to Health Adjusted Age at Death (HAAD). We do not agree that this is HALE for a subgroup of the population, because HALE considers future healthy life expectancy only. This also applies to LE (e.g., LE for a 99-year-old is about 1). 

Further, we do not agree that we should include “age of disease” or a similar phrase in the name, because one could calculate HAAD for individuals who are not affected by a condition (similar to what we did in Figure 1). 

Next, we have removed the word “expected/expectancy” from the name, because this obscures the concept of HAAD. For example, one could discuss the HAAD of disease D without knowing the expected HAAD of an individual or a group of individuals. Instead we use other words to describe the context in which HAAD is used, and introduce “aHAAD” as “average lifetime health/average HAAD”. See for example

Title: 

 “Estimating Health Adjusted Age at Death” instead of “Measuring Health Adjusted Age at Death”

Introduction: 

 “In this paper, we present a method for estimating HAAD, and show how to use the estimated HAAD to rank conditions at disease onset”

Methods:

 “The diseases have distinct properties that highlight certain characteristics of HAAD.”

 “We next set out to estimate the HAAD distribution in a population”

 “In H_D^HAAD, row number c estimates HAAD for individuals who are c years old, whereas column number d estimates HAAD for individuals who will die at age d.”

 “As opposed to the discrete AD, HAAD is continuous. For example, one individual who dies at age 80 may have HAAD=67.3, whereas another could have achieved 67.4 or 67.5.”

 “Because we do not have access to data on individuals, every individual with the same condition and the same age of onset is assumed to have the same HAAD distribution.”

 “See Figure A1 (Appendix) for details on incidence assumptions that are being used in HAAD calculations for AML and ALL in the US.”

Results:

 “Table 2 shows estimated aHAAD, T20, Q1 and Q3 for ALL, AML, schizophrenia and epilepsy in six countries”

Discussion:

 “We show how two conditions, ALL and AML, with similar estimated aHAADs have substantially different HAAD distributions.”

 “The relevance of HAAD is particularly good for preventive interventions for a disease where you are likely to capture benefits across a range of ages”

 “In this article we present HAAD as an achievement measure”

 “Understanding the underlying reasons for differences in HAAD distributions can have policy implications.”

 “It is important to note that as a measure of lifetime health among people with a specific condition, HAAD is influenced by mortality risk and morbidity from other causes, as well as by the age at which the disease occurs”

 One big problem with this paper is that it seems to use the terms YLD and dw at some places as a kind of synonyms, while these are quit different concepts.The authors should fix this problem with the paper, as this prevents me from fully understanding their calculations.

Some examples of this:

Table 1:

dw(age) = YLDAll causes(age)

How can this be? YLD is a number for the population, which will be much larger then 1, while dw is between 0 and 1.

Because of this I can also no follow the next part in the last row of the table:

YLD.All Causes(age) = 1 − (1 − dwBackground(age))(1 − YLDD(age))

Again, 1 - YLD does not make sense to me, as YLD usually is much larger than 1.

Similarly, in the text of the introduction: ”HALE measures the life expectancy in a population, adjusting for the disability experienced in the population, using age-specific mortality rates and YLDs per capit”. HALE does not use YLDs but dw’s.

 Reply: We understand that this was confusing. GBD calculates YLDs by summing dw’s over all individuals in the population. Hence, 

average dw = YLD/(population size) . 

In other words, dw is per capita YLD, and 

and 

average dw for disease D = (YLD caused by disease D)/(population with disease D) .

 Changes: We have changed the text in Table 1 and emphasized on several occasions that YLD is per capita.

 Discussion:

“Shortfall in disabilities could use the lowest YLD rates (i.e., YLD per capita) across countries as a reference for disability shortfall.”

 Table 1

“Note that YLD_(All causes) (age) is per capita.” 

“Note that YLD_D (age) is per capita.”

“To calculate the background disability, we treat YLD rates (i.e., YLD per capita) in the population as average disability weights for a given person and solve for the background disability by breaking down the all-cause disability weight into disability from the specific cause and from other causes.”

“Note that YLD_(All causes) (age) and YLD_D (age) are per capita.”

 The calculations treat rates as probabilities. This will work reasonably as long as rates are low. However, they will no longer be so at high ages. Why do you not simple use prob =1- exp(-rate*interval length) (assuming a constant rate in the interval) to calculate the probability?

 As suggested, we have changed all of our calculations using q = 1-exp(-M), where M is the mortality rate and q is the probability of dying during a year. The interval length is 1 in all our calculations. 

 We have also clarified the difference between mx and qx in Table 1:

“All cause probability of death in single-year intervals, for a total population (i.e., baseline mortality). 

From GBD 2017.

q=1-exp⁡(-M_(All causes) )

Converted from single-year MD using common demographic approximation [32].”

and 

“Probability of death due to disease and baseline mortality. These are not given directly in GBD, but can be calculated using

q_D (age)=1-exp⁡(-(M_(All causes) (age)-M_D (age)+em_D (age))) .

Substituting emD into qD yields

q_D (age)=1-exp⁡(-(M_(All causes) (age)+M_D (age)(1/(P_D (age) )-1))) .

We can see that if PD=1, meaning that all individuals in the population have a disease, qD simply becomes q. This is also the case if there is no mortality from disease, so that MD=0.” 

 Methods, line 13: The use of “health status” here is confusing, as the example only refers to being alive.

 We have changed the relevant text to: 

“For simplicity in this example, although we will use the term “lifetime health”, we disregard health/disability adjustment for time with illness and focus only on their age at death. Because there is a 99% probability that Ann will die at a younger age than Bob will, Ann’s lifetime health is lower than Bob’s in terms of total length of life (past life plus expected future life), even though Bob’s expected future life is shorter.”

 Methods, line 21: Here it is claimed that HEAD also looks at distributions, while in the example only the average (=expectancy) is used. I therefore could not follow the reasoning in this part.

 We have changed the text to: 

“HAAD enables comparison of both average lifetime health (aHAAD) and distribution of lifetime health between individuals with different diseases (e.g., at disease onset, Ann’s disease will have a very different HAAD distribution than Bob’s disease).”

 At parts the text still pertains that the metric is for individuals (while the metric is for groups of individuals). For instance in the conclusion:

“Here we have presented a method for calculating HEAD, illustrated with examples how to estimate the distribution of health across individuals”

 In principle, HAAD could be used at an individual level, but that would require information about disease history and age at death. However, as the reviewer correctly points out, the aim of this paper is to estimate HAAD for groups of individuals. Still, we focus on the estimated HAAD distribution across those individuals. 

 We have changed the relevant text: 

“Here we have presented a method for estimating lifetime health by considering HAAD, illustrated with examples how to estimate the distribution of HAAD across individuals, and shown why considering these distributions is relevant for priority setting in health care and the measurement of population health.”

 In the legend of figure 3 it is strange that PID is shorter than PIM. Although this is just a legend, it confuses.

 We have changed Fig 3 so that PID is now longer than PIM. 

Because we changed the calculations (as suggested) by using q = 1-exp(-M), the figures and tables changed. We have also edited the manuscript for clarity. 

Best wishes,

Øystein A. Haaland

---

## [Decision Letter · Decision Letter 2]

10 Jun 2020

PONE-D-19-18816R2

Estimating Health Adjusted Age at Death (HAAD)

PLOS ONE

Dear Dr. Haaland,

Thank you for submitting your manuscript to PLOS ONE. After careful consideration, we feel that it has merit but does not fully meet PLOS ONE’s publication criteria as it currently stands. Therefore, we invite you to submit a revised version of the manuscript that addresses the points raised during the review process.

A **rebuttal letter** that responds to **EACH** point raised by the academic editor and reviewer(s). You should upload this letter as a separate file labeled 'Response to Reviewers'.A **marked-up copy** of your manuscript that highlights changes made to the original version. You should upload this as a separate file labeled 'Revised Manuscript with Track Changes'.An **unmarked version** of your revised paper without tracked changes. You should upload this as a separate file labeled 'Manuscript'.

We look forward to receiving your revised manuscript.

Kind regards,

Brecht Devleesschauwer

Academic Editor

PLOS ONE

Additional Editor Comments (if provided):

The reviewer raised some final points, in particular related to the terminology used, which can be addressed in a final, minor revision round.

Reviewers' comments:

Reviewer's Responses to Questions

**Comments to the Author**

1. If the authors have adequately addressed your comments raised in a previous round of review and you feel that this manuscript is now acceptable for publication, you may indicate that here to bypass the “Comments to the Author” section, enter your conflict of interest statement in the “Confidential to Editor” section, and submit your "Accept" recommendation.

Reviewer #1: (No Response)

2. Is the manuscript technically sound, and do the data support the conclusions?

Reviewer #1: Yes

3. Has the statistical analysis been performed appropriately and rigorously? 

Reviewer #1: N/A

4. Have the authors made all data underlying the findings in their manuscript fully available?

Reviewer #1: (No Response)

5. Is the manuscript presented in an intelligible fashion and written in standard English?

Reviewer #1: Yes

6. Review Comments to the Author

Reviewer #1: The authors have achieved a lot with respect to clarification and use of terminology since the previous revision.

Main remaining point to be addressed:

The use of disability weight (dw) to stand for disability as a quantity with a time dimension will be very confusing to the reader familiar with 'dw' as an elicited weight (not something that can be seen as a population average, or that necessarily can be decomposed).

In Table 1, the authors describe the calculation of "... the background disability, we treat YLD rates (i.e., YLD per capita) in the population as average disability weights for a given person and solve for the background disability by breaking down the all-cause disability weight into disability

from the specific cause and from other causes.

I would ask the authors to not conflate disability weight and YLD rate, and so choose terms that do not overlap with 'standard' burden of disease terminology.

7. PLOS authors have the option to publish the peer review history of their article (what does this mean?). If published, this will include your full peer review and any attached files.

Reviewer #1: No

---

## [Author Response · Author response to Decision Letter 2]

13 Jun 2020

Dear editor

Please find our replies to the reviewers below. 

Reviewer #1: 

Comment: 

“The authors have achieved a lot with respect to clarification and use of terminology since the previous revision.

Main remaining point to be addressed:

The use of disability weight (dw) to stand for disability as a quantity with a time dimension will be very confusing to the reader familiar with 'dw' as an elicited weight (not something that can be seen as a population average, or that necessarily can be decomposed).

In Table 1, the authors describe the calculation of "... the background disability, we treat YLD rates (i.e., YLD per capita) in the population as average disability weights for a given person and solve for the background disability by breaking down the all-cause disability weight into disability from the specific cause and from other causes.

I would ask the authors to not conflate disability weight and YLD rate, and so choose terms that do not overlap with 'standard' burden of disease terminology.”

Reply: 

Although we understand the concern of the reviewer, according to the GBD, “YLDs were estimated as the product of prevalence estimate and a disability weight for health states […]” (Lancet 2018; 392: 1789–858). In Table 1 this corresponds to: 

dw_D (age)=(YLD_D (age))/(P_D (age)) ,

and the rest of the derivations follow from this. This terminology may be confusing, but we worry that it will be even more confusing if we introduce new terminology do describe the relationship between YLDs and disability weights. Therefore, we prefer to keep the terms YLD and “disability weight” as they are. 

To explain the GBD approach, the Introduction says: 

“GBD uses a prevalence-based approach, where DALYs are calculated for a set of diseases by summing the years of life lost (YLL) compared to a reference life expectancy and years lived with disability (YLD) in one particular year due to each disease [19]. For a particular condition and a particular year, YLL is the sum of all the years lost for the individuals who died from the condition during that year. The reference is the age-adjusted life expectancy (LE) from a life table derived from the mortality rates in the locations with the lowest age-specific mortality in the GBD study [20]. YLD, on the other hand, is the sum of the health loss due to the condition during the year across people living with the condition [21]. DALYs aggregated from YLLs and YLDs are a measure of overall population burden. HALE measures the life expectancy in a population, adjusting for the disability experienced in the population, using age-specific mortality rates and YLDs per capita [22].”

In Table 1 the definition of YLD_D now reads:

“Years Lived with Disability (per capita in one year) of disease in 5-year age intervals. 

From GBD 2017.

Assumed to be the same in all 1-year intervals.”

We hope that you find this line of reasoning convincing. 

Best wishes,

Øystein A. Haaland

---

## [Editor Report · Decision Letter 3]

26 Jun 2020

Estimating Health Adjusted Age at Death (HAAD)

PONE-D-19-18816R3

Dear Dr. Haaland,

We’re pleased to inform you that your manuscript has been judged scientifically suitable for publication and will be formally accepted for publication once it meets all outstanding technical requirements.

Kind regards,

Brecht Devleesschauwer

Academic Editor

PLOS ONE
---

## [Editor Report · Acceptance letter]

30 Jun 2020

PONE-D-19-18816R3 

Estimating Health Adjusted Age at Death (HAAD) 

Dear Dr. Haaland:

I'm pleased to inform you that your manuscript has been deemed suitable for publication in PLOS ONE. Congratulations! Your manuscript is now with our production department. 

Kind regards, 

on behalf of

Prof. Dr. Brecht Devleesschauwer 

Academic Editor

PLOS ONE